# Learning Embeddings into Entropic Wasserstein Spaces

**Charlie Frogner**
MIT CSAIL and MIT-IBM Watson AI Lab
`frogner@mit.edu`

**Farzaneh Mirzazadeh**
MIT-IBM Watson AI Lab and IBM Research
`farzaneh@ibm.com`

**Justin Solomon**
MIT CSAIL and MIT-IBM Watson AI Lab
`jsolomon@mit.edu`

## Abstract

Euclidean embeddings of data are fundamentally limited in their ability to capture latent semantic structures, which need not conform to Euclidean spatial assumptions. Here we consider an alternative, which embeds data as discrete probability distributions in a Wasserstein space, endowed with an optimal transport metric. Wasserstein spaces are much larger and more flexible than Euclidean spaces, in that they can successfully embed a wider variety of metric structures. We exploit this flexibility by learning an embedding that captures semantic information in the Wasserstein distance between embedded distributions. We examine empirically the representational capacity of our learned Wasserstein embeddings, showing that they can embed a wide variety of metric structures with smaller distortion than an equivalent Euclidean embedding. We also investigate an application to word embedding, demonstrating a unique advantage of Wasserstein embeddings: We can visualize the high-dimensional embedding directly, since it is a probability distribution on a low-dimensional space. This obviates the need for dimensionality reduction techniques like t-SNE for visualization.

## 1 Introduction

Learned embeddings form the basis for many state-of-the-art learning systems. Word embeddings like word2vec (Mikolov et al., 2013), GloVe (Pennington et al., 2014), fastText (Bojanowski et al., 2017), and ELMo (Peters et al., 2018) are ubiquitous in natural language processing, where they are used for tasks like machine translation (Neubig et al., 2018), while graph embeddings (Nickel et al., 2016) like node2vec (Grover & Leskovec, 2016) are used to represent knowledge graphs and pre-trained image models (Simon et al., 2016) appear in many computer vision pipelines.

An effective embedding should capture the semantic structure of the data with high fidelity, in a way that is amenable to downstream tasks. This makes the choice of a target space for the embedding important, since different spaces can represent different types of semantic structure. The most common choice is to embed data into Euclidean space, where distances and angles between vectors encode their levels of association (Mikolov et al., 2013; Weston et al., 2011; Kiros et al., 2014; Mirzazadeh et al., 2014). Euclidean spaces, however, are limited in their ability to represent complex relationships between inputs, since they make restrictive assumptions about neighborhood sizes and connectivity. This drawback has been documented recently for tree-structured data, for example, where spaces of negative curvature are required due to exponential scaling of neighborhood sizes (Nickel & Kiela, 2017; 2018).

In this paper, we embed input data as probability distributions in a Wasserstein space. Wasserstein spaces endow probability distributions with an *optimal transport* metric, which measures the distance traveled in transporting the mass in one distribution to match another. Recent theory has shown that Wasserstein spaces are quite flexible—more so than Euclidean spaces—allowing a variety of other metric spaces to be embedded within them while preserving their original distance metrics. As

such, they make attractive targets for embeddings in machine learning, where this flexibility might capture complex relationships between objects when other embeddings fail to do so.

Unlike prior work on Wasserstein embeddings, which has focused on embedding into Gaussian distributions (Muzellec & Cuturi, 2018; Zhu et al., 2018), we embed input data as discrete distributions supported at a fixed number of points. In doing so, we attempt to access the full flexibility of Wasserstein spaces to represent a wide variety of structures.

Optimal transport metrics and their gradients are costly to compute, requiring the solution of a linear program. For efficiency, we use an approximation to the Wasserstein distance called the Sinkhorn divergence (Cuturi, 2013), in which the underlying transport problem is regularized to make it more tractable. While less well-characterized theoretically with respect to embedding capacity, the Sinkhorn divergence is computed efficiently by a fixed-point iteration. Moreover, recent work has shown that it is suitable for gradient-based optimization via automatic differentiation (Genevay et al., 2018b). To our knowledge, our work is the first to explore embedding properties of the Sinkhorn divergence.

We empirically investigate two settings for Wasserstein embeddings. First, we demonstrate their representational capacity by embedding a variety of complex networks, for which Wasserstein embeddings achieve higher fidelity than both Euclidean and hyperbolic embeddings. Second, we compute Wasserstein word embeddings, which show retrieval performance comparable to existing methods. One major benefit of our embedding is that the distributions can be visualized directly, unlike most embeddings, which require a dimensionality reduction step such as t-SNE before visualization. We demonstrate the power of this approach by visualizing the learned word embeddings.

## 2 PRELIMINARIES

### 2.1 OPTIMAL TRANSPORT AND WASSERSTEIN DISTANCE

The $p$-**Wasserstein distance** between probability distributions $\mu$ and $\nu$ over a metric space $\mathcal{X}$ is

$$\mathcal{W}_p(\mu, \nu) = \left( \inf_{\pi \in \Pi(\mu,\nu)} \int_{\mathcal{X} \times \mathcal{X}} d(x_1, x_2)^p \, d\pi(x_1, x_2) \right)^{\frac{1}{p}}, \tag{1}$$

where the infimum is taken over **transport plans** $\pi$ that distribute the mass in $\mu$ to match that in $\nu$, with the $p$-th power of the **ground metric** $d(x_1, x_2)$ on $\mathcal{X}$ giving the cost of moving a unit of mass from support point $x_1 \in \mathcal{X}$ underlying distribution $\mu$ to point $x_2 \in \mathcal{X}$ underlying $\nu$. The Wasserstein distance is the cost of the optimal transport plan matching $\mu$ and $\nu$ (Villani, 2003).

In this paper, we are concerned with **discrete distributions** supported on finite sets of points in $\mathbb{R}^n$:

$$\mu = \sum_{i=1}^{M} \mathbf{u}_i \delta_{\mathbf{x}^{(i)}} \qquad \text{and} \qquad \nu = \sum_{i=1}^{N} \mathbf{v}_i \delta_{\mathbf{y}^{(i)}}. \tag{2}$$

Here, $\mathbf{u}$ and $\mathbf{v}$ are vectors of nonnegative weights summing to 1, and $\{\mathbf{x}^{(i)}\}_{i=1}^{M}, \{\mathbf{y}^{(i)}\}_{i=1}^{N} \subset \mathbb{R}^n$ are the **support points**. In this case, the transport plan $\pi$ matching $\mu$ and $\nu$ in Equation 1 becomes discrete as well, supported on the product of the two support sets. Define $D \in \mathbb{R}_+^{M \times N}$ to be the matrix of pairwise ground metric distances, with $D_{ij} = d(\mathbf{x}^{(i)}, \mathbf{y}^{(j)})$. Then, for discrete distributions, Equation 1 is equivalent to solving the following:

$$\mathcal{W}_p(\mu, \nu)^p = \min_{T \geq 0} \operatorname{tr}(D^p T^\top) \quad \text{subject to} \quad T\mathbf{1} = \boldsymbol{u}, \quad T^\top \mathbf{1} = \boldsymbol{v}, \tag{3}$$

with $T_{ij}$ giving the transported mass between $\mathbf{x}_i$ and $\mathbf{y}_j$. The power $D^p$ is taken elementwise.

### 2.2 SINKHORN DIVERGENCE

Equation 3 is a linear program that can be challenging to solve in practice. To improve efficiency, recent learning algorithms use an entropic regularizer proposed by Cuturi (2013). The resulting **Sinkhorn divergence** solves a modified version of Equation 3:

$$\mathcal{W}_p^\lambda(\mu, \nu)^p = \min_{T \geq 0} \operatorname{tr}(D^p T^\top) + \lambda \operatorname{tr}\left( T(\log(T) - \mathbf{1}\mathbf{1}^\top)^\top \right) \quad \text{s.t. } T\mathbf{1} = \boldsymbol{u}, \ T^\top \mathbf{1} = \boldsymbol{v}, \tag{4}$$

where $\log(\cdot)$ is applied elementwise and $\lambda \geq 0$ is the regularization parameter. For $\lambda > 0$, the optimal solution takes the form $T^* = \Delta(\boldsymbol{r}) \exp\left(-D^p/\lambda\right) \Delta(\boldsymbol{c})$, where $\Delta(\boldsymbol{r})$ and $\Delta(\boldsymbol{c})$ are diagonal matrices with diagonal elements $\boldsymbol{r}$ and $\boldsymbol{c}$, resp. Rather than optimizing over matrices $T$, one can optimize for $\boldsymbol{r}$ and $\boldsymbol{c}$, reducing the size of the problem to $M + N$. This can be solved via *matrix balancing*, starting from an initial matrix $K := \exp(\frac{-D^p}{\lambda})$ and alternately projecting onto the marginal constraints until convergence:

$$\boldsymbol{r} \leftarrow \boldsymbol{u}./K\boldsymbol{c} \qquad \boldsymbol{c} \leftarrow \boldsymbol{v}./K^\top \boldsymbol{r}. \qquad (5)$$

Here, $./$ denotes elementwise division for vectors.

Beyond simplicity of implementation, Equation 5 has an additional advantage for machine learning: The steps of this algorithm are differentiable. With this observation in mind, Genevay et al. (2018b) incorporate entropic transport into learning pipelines by applying automatic differentiation (back propagation) to a fixed number of Sinkhorn iterations.

## 2.3 What can we embed in theory?

Given two metric spaces $\mathcal{A}$ and $\mathcal{B}$, an embedding of $\mathcal{A}$ into $\mathcal{B}$ is a map $\phi : \mathcal{A} \to \mathcal{B}$ that approximately preserves distances, in the sense that the distortion is small:

$$Ld_{\mathcal{A}}(u, v) \leq d_{\mathcal{B}}(\phi(u), \phi(v)) \leq CLd_{\mathcal{A}}(u, v), \quad \forall u, v \in \mathcal{A}, \qquad (6)$$

for some uniform constants $L > 0$ and $C \geq 1$. The distortion of the embedding $\phi$ is the smallest $C$ such that Equation 6 holds.

One can characterize how "large" a space is (its *representational capacity*) by the spaces that embed into it with low distortion. In practical terms, this capacity determines the types of data (and relationships between them) that can be well-represented in the embedding space. $\mathbb{R}^n$ with the Euclidean metric, for example, embeds into the $L^1$ metric with low distortion, while the reverse is not true (Deza & Laurent, 2009). We do not expect Manhattan-structured data to be well-represented in Euclidean space, no matter how clever the mapping.

Wasserstein spaces are very large: Many spaces can embed into Wasserstein spaces with low distortion, even when the converse is not true. $\mathcal{W}_p(\mathcal{A})$, for $\mathcal{A}$ an arbitrary metric space, embeds any product space $\mathcal{A}^n$, for example (Kloeckner, 2010), via discrete distributions supported at $n$ points. Even more generally, certain Wasserstein spaces are **universal**, in the sense that they can embed arbitrary metrics on finite spaces. $\mathcal{W}_1(\ell^1)$ is one such space (Bourgain, 1986), and it is still an open problem to determine if $\mathcal{W}_1(\mathbb{R}^k)$ is universal for any $k < +\infty$. Recently it has been shown that every finite metric space embeds the $\frac{1}{p}$ power of its metric into $\mathcal{W}_p(\mathbb{R}^3)$, $p > 1$, with vanishing distortion (Andoni et al., 2015). A hopeful interpretation suggests that $\mathcal{W}_1(\mathbb{R}^3)$ may be a plausible target space for arbitrary metrics on symbolic data, with a finite set of symbols; we are unaware of similar universality results for $L^p$ or hyperbolic spaces, for example.

The reverse direction, embedding Wasserstein spaces into others, is well-studied in the case of discrete distributions. Theoretical results in this domain are motivated by interest in efficient algorithms for approximating Wasserstein distances by embedding into spaces with easily-computed metrics. In this direction, low-distortion embeddings are difficult to find. $\mathcal{W}_2(\mathbb{R}^3)$, for example, is known not to embed into $L^1$ (Andoni et al., 2016). Some positive results exist, nevertheless. For a Euclidean ground metric, for example, the 1-Wasserstein distance can be approximated in a wavelet domain (Shirdhonkar & Jacobs, 2008) or by high-dimensional embedding into $L^1$ (Indyk & Thaper, 2003).

In §4, we empirically investigate the embedding capacity of Wasserstein spaces, by attempting to learn low-distortion embeddings for a variety of input spaces. For efficiency, we replace the Wasserstein distance by its entropically-regularized counterpart, the Sinkhorn divergence (§2.2). The embedding capacity of Sinkhorn divergences is previously unstudied, to our knowledge, except in the weak sense that the approximation error with respect to the Wasserstein distance vanishes with the regularizer taken to zero (Carlier et al., 2017; Genevay et al., 2018a).

## 2.4 Related work

While learned vector space embeddings have a long history, there is a recent trend in the representation learning community towards more complex target spaces, such as spaces of probability distributions (Vilnis & McCallum, 2015; Athiwaratkun & Wilson, 2018), Euclidean norm balls (Mirzazadeh

et al., 2015; Mirzazadeh, 2017), Poincaré balls (Nickel & Kiela, 2017), and Riemannian manifolds (Nickel & Kiela, 2018). From a modeling perspective, these more complex structures assist in representing uncertainty about the objects being embedded (Vilnis & McCallum, 2015; Bojchevski & Günnemann, 2018) as well as complex relations such as inclusion, exclusion, hierarchy, and ordering (Mirzazadeh et al., 2015; Vendrov et al., 2015; Athiwaratkun & Wilson, 2018). In the same vein, our work takes probability distributions in a Wasserstein space as our embedding targets.

The distance or discrepancy measure between target structures is a major defining factor for a representation learning model. $L_p$ distances as well as angle-based discrepancies are fairly common (Mikolov et al., 2013), as is the KL divergence (Kullback & Leibler, 1951), when embedding into probability distributions. For distributions, however, the KL divergence and $L_p$ distances are problematic, in the sense that they ignore the geometry of the domain of the distributions being compared. For distributions with disjoint support, for example, these divergences do not depend on the separation between the supports. Optimal transport distances (Villani, 2008; Cuturi, 2013; Peyré & Cuturi, 2017; Solomon, 2018), on the other hand, explicitly account for the geometry of the domain. Hence, models based on optimal transport are gaining popularity in machine learning; see (Rubner et al., 1998; Courty et al., 2014; Frogner et al., 2015; Kusner et al., 2015; Arjovsky et al., 2017; Genevay et al., 2018b; Claici et al., 2018; Singh et al., 2018) for some examples.

Learned embeddings into Wasserstein spaces are relatively unexplored. Recent research proposes embedding into Gaussian distributions (Muzellec & Cuturi, 2018; Zhu et al., 2018). Restricting to parametric distributions enables closed-form expressions for transport distances, but the resulting representation space may lose expressiveness. We note that Courty et al. (2018) study embedding in the opposite direction, from Wasserstein into Euclidean space. In contrast, we learn to embed into the space of discrete probability distributions endowed with the Wasserstein distance. Discrete distributions are dense in $\mathcal{W}_2$ (Kloeckner, 2012; Brancolini et al., 2009).

# 3 LEARNING WASSERSTEIN EMBEDDINGS

## 3.1 THE LEARNING PROBLEM

We consider the task of recovering a pairwise distance or similarity relationship that may be only partially observed. We are given a collection of objects $\mathcal{C}$—these can be words, symbols, images, or any other data—as well as samples $\left\{ \left( u^{(i)}, v^{(i)}, r(u^{(i)}, v^{(i)}) \right) \right\}$ of a target relationship $r : \mathcal{C} \times \mathcal{C} \to \mathbb{R}$ that tells us the degree to which pairs of objects are related.

Our objective is to find a map $\phi : \mathcal{C} \to \mathcal{W}_p(\mathcal{X})$ such that the relationship $r(u, v)$ can be recovered from the Wasserstein distance between $\phi(u)$ and $\phi(v)$, for any $u, v \in \mathcal{C}$. Examples include:

1. METRIC EMBEDDING: $r$ is a distance metric, and we want $\mathcal{W}_p(\phi(u), \phi(v)) \approx r(u, v)$ for all $u, v \in \mathcal{C}$.

2. GRAPH EMBEDDING: $\mathcal{C}$ contains the vertices of a graph and $r : \mathcal{C} \times \mathcal{C} \to \{0, 1\}$ is the adjacency relation; we would like the neighborhood of each $\phi(u)$ in $\mathcal{W}_p$ to coincide with graph adjacency.

3. WORD EMBEDDING: $\mathcal{C}$ contains individual words and $r$ is a semantic similarity between words. We want distances in $\mathcal{W}_p$ to predict this semantic similarity.

Although the details of each task require some adjustment to the learning architecture, our basic representation and training procedure detailed below applies to all three examples.

## 3.2 OPTIMIZATION

Given a set of training samples $\mathcal{S} = \left\{ \left( u^{(i)}, v^{(i)}, r^{(i)} \right) \right\}_{i=1}^{N} \subset \mathcal{C} \times \mathcal{C} \times \mathbb{R}$, we want to learn a map $\phi : \mathcal{C} \to \mathcal{W}_p(\mathcal{X})$. We must address two issues.

First we must define the range of our map $\phi$. The whole of $\mathcal{W}_p(\mathcal{X})$ is infinite-dimensional, and for a tractable problem we need a finite-dimensional output. We restrict ourselves to discrete distributions with an *a priori* fixed number of support points $M$, reducing optimal transport to the linear program in Equation 3. Such a distribution is parameterized by the locations of its support points $\{\mathbf{x}^{(j)}\}_{j=1}^{M}$, forming a point cloud in the ground metric space $\mathcal{X}$. For simplicity, we restrict to uniform weights $\mathbf{u}, \mathbf{v} \propto \mathbf{1}$, although it is possible to optimize simultaneously over weights and locations. As noted

in (Brancolini et al., 2009; Kloeckner, 2012; Claici et al., 2018), however, when constructing a discrete $M$-point approximation to a fixed target distribution, allowing non-uniform weights does not improve the asymptotic approximation error.[1]

The second issue is that, as noted in §2.2, exact computation of $\mathcal{W}_p$ in general is costly, requiring the solution of a linear program. As in (Genevay et al., 2018b), we replace $\mathcal{W}_p$ with the Sinkhorn divergence $\mathcal{W}_p^\lambda$, which is solvable by a the fixed-point iteration in Equation 5. Learning then takes the form of empirical loss minimization:

$$\phi_* = \arg\min_{\phi \in \mathcal{H}} \frac{1}{N} \sum_{i=1}^{N} \mathcal{L}\left(\mathcal{W}_p^\lambda\left(\phi(u^{(i)}), \phi(v^{(i)})\right), r^{(i)}\right), \tag{7}$$

over a hypothesis space of maps $\mathcal{H}$. The loss $\mathcal{L}$ is problem-specific and scores the similarity between the regularized Wasserstein distance $\mathcal{W}_p^\lambda$ and the target relationship $r$ at $\left(u^{(i)}, v^{(i)}\right)$. As mentioned in §2.2, gradients are available from automatic differentiation of the Sinkhorn procedure, and hence with a suitable loss function the learning objective Equation 7 can be optimized by standard gradient-based methods. In our experiments, we use the Adam optimizer (Kingma & Ba, 2014).

# 4 EMPIRICAL STUDY

## 4.1 REPRESENTATIONAL CAPACITY: EMBEDDING COMPLEX NETWORKS

We first demonstrate the representational power of learned Wasserstein embeddings. As discussed in §2.3, theory suggests that Wasserstein spaces are quite flexible, in that they can embed a wide variety of metrics with low distortion. We show that this is true in practice as well.

To generate a variety of metrics to embed, we take networks with various patterns of connectivity and compute the shortest-path distances between vertices. The collection of vertices for each network serves as the input space $\mathcal{C}$ for our embedding, and our goal is to learn a map $\phi : \mathcal{C} \to \mathcal{W}_p(\mathbb{R}^k)$ such that the 1-Wasserstein distance $\mathcal{W}_1(\phi(u), \phi(v))$ matches as closely as possible the shortest path distance between vertices $u$ and $v$, for all pairs of vertices. We learn a **minimum-distortion embedding**: Given a fully-observed distance metric $d_{\mathcal{C}} : \mathcal{C} \times \mathcal{C} \to \mathbb{R}$ in the input space, we minimize the mean distortion:

$$\phi_* = \arg\min_{\phi} \frac{1}{\binom{n}{2}} \sum_{j>i} \frac{|\mathcal{W}_1^\lambda(\phi(v_i), \phi(v_j)) - d_{\mathcal{C}}(v_i, v_j)|}{d_{\mathcal{C}}(v_i, v_j)}. \tag{8}$$

$\phi$ is parameterized as in §3.2, directly specifying the support points of the output distribution.

We examine the performance of Wasserstein embedding using both random networks and real networks. The random networks in particular allow us systematically to test robustness of the Wasserstein embedding to particular properties of the metric we are attempting to embed. Note that these experiments do not explore generalization performance: We are purely concerned with the representational capacity of the learned Wasserstein embeddings.

For random networks, we use three standard generative models: Barabási–Albert (Albert & Barabási, 2002), Watts–Strogatz (Watts & Strogatz, 1998), and the stochastic block model (Holland et al., 1983). **Random scale-free networks** are generated from the Barabási–Albert model, and possess the property that distances are on average much shorter than in a Euclidean spatial graph, scaling like the logarithm of the number of vertices. **Random small-world networks** are generated from the Watts–Strogatz model; in addition to log-scaling of the average path length, the vertices of Watts–Strogatz graphs cluster into distinct neighborhoods. **Random community-structured networks** are generated from the stochastic block model, which places vertices within densely-connected communities, with sparse connections between the different communities. We additionally generate **random trees** by choosing a random number of children[2] for each node, progressing in breadth-first order until a specified total number of nodes is reached. In all cases, we generate networks with 128 vertices.

---

[1]In both the non-uniform and uniform cases, the order of convergence in $\mathcal{W}_p$ of the nearest weighted point cloud to the target measure, as we add more points, is $\mathcal{O}(M^{-1/d})$, for a $d$-dimensional ground metric space. This assumes the underlying measure is absolutely continuous and compactly-supported.

[2]Each non-leaf node has a number of children drawn uniformly from $\{2, 3, 4\}$.

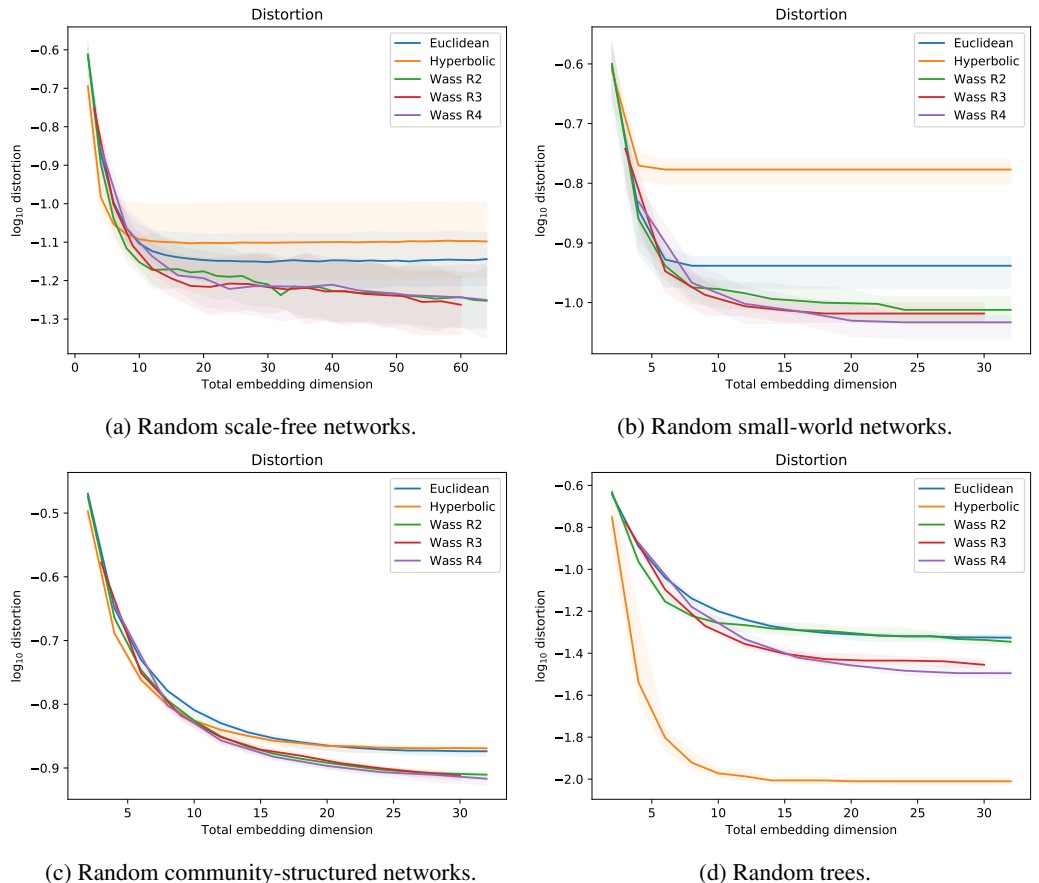

(a) Random scale-free networks.

(b) Random small-world networks.

(c) Random community-structured networks.

(d) Random trees.

Figure 1: Random networks: Learned Wasserstein embeddings achieve lower distortion than Euclidean/hyperbolic embeddings. Hyperbolic embeddings outperform specifically on random trees.

We compare against two baselines, trained using the same distortion criterion and optimization method: **Euclidean embeddings**, and **hyperbolic embeddings**. Euclidean embeddings we expect to perform poorly on all of the chosen graph types, since they are limited to spatial relationships with zero curvature. Hyperbolic embeddings model tree-structured metrics, capturing the exponential scaling of graph neighborhoods; they have been suggested for a variety of other graph families as well (Zhao et al., 2011).

Figure 1 shows the result of embedding random networks.[3] As the total embedding dimension increases, the distortion decreases for all methods. Importantly, Wasserstein embeddings achieve lower distortion than Euclidean and hyperbolic embeddings, establishing their flexibility under the varying conditions represented by the different network models. In some cases, the Wasserstein distortion continues to decrease long after the other embeddings have saturated their capacity. As expected, hyperbolic space significantly outperforms both Euclidean and Wasserstein specifically on tree-structured metrics.

We test $\mathbb{R}^2$, $\mathbb{R}^3$, and $\mathbb{R}^4$ as ground metric spaces. For all of the random networks we examined, the performance between $\mathbb{R}^3$ and $\mathbb{R}^4$ is nearly indistinguishable. This observation is consistent with theoretical results (§2.3) suggesting that $\mathbb{R}^3$ is sufficient to embed a wide variety of metrics.

We also examine fragments of real networks: an ArXiv co-authorship network, an Amazon product co-purchasing network, and a Google web graph (Leskovec & Krevl, 2014). For each graph fragment, we choose uniformly at random a starting vertex and then extract the subgraph on 128 vertices taken in breadth-first order from that starting vertex. Distortion results are shown in Figure 2. Again the Wasserstein embeddings achieve lower distortion than Euclidean or hyperbolic embeddings.

---

[3]The solid line is the median over 20 randomly-generated inputs, while shaded is the middle 95%.

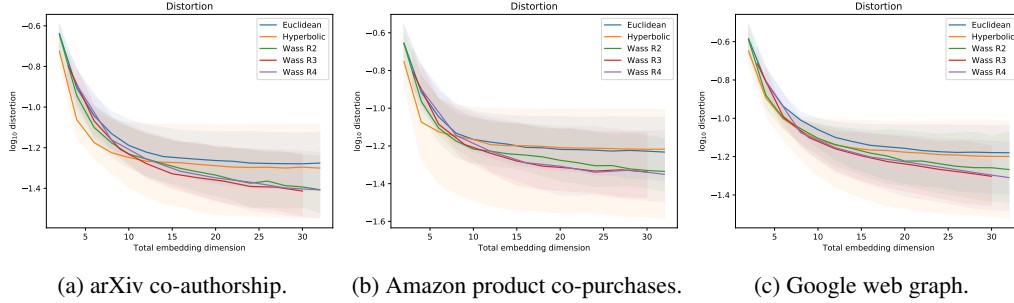

Figure 2: Real networks: Learned Wasserstein embeddings achieve lower distortion than Euclidean and hyperbolic embeddings of real network fragments.

| | | |
|---|---|---|
| $\mathcal{W}_1^\lambda(\mathbb{R}^2)$ | one: | f, two, i, after, four |
| | united: | series, professional, team, east, central |
| | algebra: | skin, specified, equation, hilbert, reducing |
| $\mathcal{W}_1^\lambda(\mathbb{R}^3)$ | one: | two, three, s, four, after |
| | united: | kingdom, australia, official, justice, officially |
| | algebra: | binary, distributions, reviews, ear, combination |
| $\mathcal{W}_1^\lambda(\mathbb{R}^4)$ | one: | six, eight, zero, two, three |
| | united: | army, union, era, treaty, federal |
| | algebra: | tables, transform, equations, infinite, differential |

Table 1: Change in the 5-nearest neighbors when increasing dimensionality of each point cloud with fixed total length of representation.

## 4.2 WORD2CLOUD: WASSERSTEIN WORD EMBEDDINGS

In this section, we embed words as point clouds. In a sentence $s = (\mathbf{x}_0, \ldots, \mathbf{x}_n)$, a word $\mathbf{x}_i$ is associated with word $\mathbf{x}_j$ if $\mathbf{x}_j$ is in the *context* of $\mathbf{x}_i$, which is a symmetric window around $\mathbf{x}_i$. This association is encoded as a label $r$; $r_{\mathbf{x}_i, \mathbf{x}_j} = 1$ if and only if $|i - j| \leq l$ where $l$ is the window size. For word embedding, we use a contrastive loss function (Hadsell et al., 2006)

$$\phi_* = \arg\min_\phi \sum_s \sum_{\mathbf{x}_i, \mathbf{x}_j \in s} r_{\mathbf{x}_i, \mathbf{x}_j} \left( \mathcal{W}_1^\lambda \big( \phi(\mathbf{x}_i), \phi(\mathbf{x}_j) \big) \right)^2 + (1 - r_{\mathbf{x}_i, \mathbf{x}_j}) \left( \left[ m - \mathcal{W}_1^\lambda \big( \phi(\mathbf{x}_i), \phi(\mathbf{x}_j) \big) \right]_+ \right)^2,$$

(9)

which tries to embed words $\mathbf{x}_i, \mathbf{x}_j$ near each other in terms of 1-Wasserstein distance (here $\mathcal{W}_1^\lambda$) if they co-occur in the context; otherwise, it prefers moving them at least distance $m$ away from one another. This approach is similar to that suggested by Mikolov et al. (2013), up to the loss and distance functions.

We use a Siamese architecture (Bromley et al., 1993) for our model, with negative sampling (as in Mikolov et al. (2013)) for selecting words outside the context. The network architecture in each branch consists of a linear layer with $64$ nodes followed by our point cloud embedding layer. The two branches of the Siamese network connect via the Wasserstein distance, computed as in §2.2. The training dataset is Text8[4], which consists of a corpus with $17M$ tokens from Wikipedia and is commonly used as a language modeling benchmark. We choose a vocabulary of 8000 words and a context window size of $l = 2$ (i.e., 2 words on each side), $\lambda = 0.05$, number of epochs of 3, negative sampling rate of 1 per positive and Adam (Kingma & Ba, 2014) for optimization.

We first study the effect of dimensionality of the point cloud on the quality of the semantic neighborhood captured by the embedding. We fix the total number of output parameters, being the product of the number of support points and the dimension of the support space, to 63 or 64 parameters. Table 1 shows the 5 nearest neighbors in the embedding space. Notably, increasing the dimensionality directly improves the quality of the learned representation. Interestingly, it is more effective to use a budget of 64 parameters in a 16-point, 4-dimensional cloud than in a 32-point, 2-dimensional cloud.

Next we evaluate these models on a number of benchmark retrieval tasks from (Faruqui & Dyer, 2014), which score a method by the correlation of its output similarity scores with human similarity

---

[4]From http://mattmahoney.net/dc/text8.zip

| Task Name | # Pairs | $\mathcal{W}_1^\lambda(\mathbb{R}^2)$ 17M | $\mathcal{W}_1^\lambda(\mathbb{R}^3)$ 17M | $\mathcal{W}_1^\lambda(\mathbb{R}^4)$ 17M | R — | M 63M | S 631M | G 900M | W 100B |
|---|---|---|---|---|---|---|---|---|---|
| RG-65 | 65 | 0.32 | 0.67 | 0.81 | 0.27 | -0.02 | 0.50 | 0.66 | 0.54 |
| WS-353 | 353 | 0.15 | 0.27 | 0.33 | 0.24 | 0.10 | 0.49 | 0.62 | 0.64 |
| WS-353-S | 203 | 0.23 | 0.40 | 0.44 | 0.36 | 0.15 | 0.61 | 0.70 | 0.70 |
| WS-353-R | 252 | 0.05 | 0.19 | 0.21 | 0.18 | 0.09 | 0.40 | 0.56 | 0.61 |
| MC-30 | 30 | 0.04 | 0.45 | 0.54 | 0.47 | -0.14 | 0.57 | 0.66 | 0.63 |
| Rare-Word | 2034 | 0.06 | 0.22 | 0.10 | 0.29 | 0.11 | 0.39 | 0.06 | 0.39 |
| MEN | 3000 | 0.25 | 0.28 | 0.26 | 0.24 | 0.09 | 0.57 | 0.31 | 0.65 |
| MTurk-287 | 287 | 0.40 | 0.38 | 0.49 | 0.33 | 0.09 | 0.59 | 0.36 | 0.67 |
| MTurk-771 | 771 | 0.11 | 0.23 | 0.25 | 0.26 | 0.10 | 0.50 | 0.32 | 0.57 |
| SimLex-999 | 999 | 0.09 | 0.05 | 0.07 | 0.23 | 0.01 | 0.27 | 0.10 | 0.31 |
| Verb-143 | 144 | 0.03 | 0.03 | 0.16 | 0.29 | 0.06 | 0.36 | 0.44 | 0.27 |

Table 2: Performance on a number of similarity benchmarks when dimensionality of point clouds increase given a fixed total number of parameters. The middle block shows the performance of the proposed models. The right block shows the performance of baselines. The training corpus size when known appears below each model name.

judgments, for various pairs of words. Results are shown in Table 2. The results of our method, which use Sinkhorn distance to compute the point cloud (dis)similarities, appear in the middle block of Table 2. Again, we mainly see gradual improvement with increasing dimensionality of the point clouds. The right block in Table 2 shows baselines: Respectively, RNN(80D) (Kombrink et al., 2011), Metaoptimize (50D) (Turian et al., 2010), SENNA (50D) (Collobert, 2011) Global Context (50D) (Huang et al., 2012) and word2vec (80D) (Mikolov et al., 2013). In the right block, as in (Faruqui & Dyer, 2014), the cosine similarity is used for point embeddings. The reported performance measure is the correlation with ground-truth rankings, computed as in (Faruqui & Dyer, 2014). Note there are many ways to improve the performance: increasing the vocabulary/window size/number of epochs/negative sampling rate, using larger texts, and accelerating performance. We defer this tuning to future work focused specifically on NLP.

### 4.2.1 DIRECT, INTERPRETABLE VISUALIZATION OF HIGH-DIMENSIONAL EMBEDDINGS

Wasserstein embeddings over low-dimensional ground metric spaces have a unique property: We can **directly visualize the embedding**, which is a point cloud in the low-dimensional ground space. This is not true for most existing embedding methods, which rely on dimensionality reduction techniques such as t-SNE for visualization. Whereas dimensionality reduction only approximately captures proximity of points in the embedding space, with Wasserstein embeddings we can display the exact embedding of each input, by visualizing the point cloud.

We demonstrate this property by visualizing the learned word representations. Importantly, each point cloud is strongly clustered, which leads to apparent, distinct modes in its density. We therefore use kernel density estimation to visualize the densities. In Figure 3a, we visualize three distinct words, thresholding each density at a low value and showing its upper level set to reveal the modes. These level sets are overlaid, with each color in the figure corresponding to a distinct embedded word. The density for each word is depicted by the opacity of the color within each level set.

It is easy to visualize multiple sets of words in aggregate, by assigning all words in a set a single color. This immediately reveals how well-separated the sets are, as shown in Figure 3b: As expected, military and political terms overlap, while names of sports are more distant.

Examining the embeddings in more detail, we can dissect relationships (and confusion) between different sets of words. We observe that each word tends to concentrate its mass in two or more distinct regions. This multimodal shape allows for multifaceted relationships between words, since a word can partially overlap with many distinct groups of words simultaneously. Figure 3c shows the embedding for a word that has multiple distinct meanings (kind), alongside synonyms for both senses of the word (nice, friendly, type). We see that kind has two primary modes, which overlap separately with friendly and type. nice is included to show a failure of the embedding to capture the full semantics: Figure 3d shows that the network has learned that nice is a city in France, ignoring its interpretation as an adjective. This demonstrates the potential of this visualization for debugging, helping identify and attribute an error.

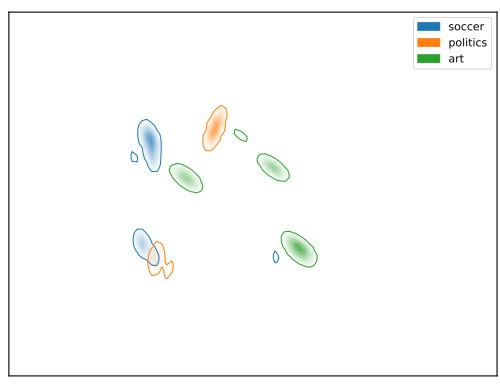

(a) Densities of three embedded words.

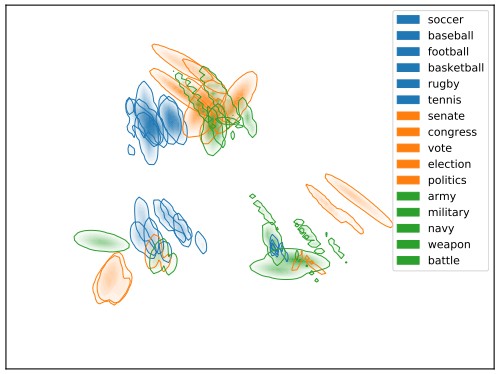
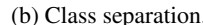

(b) Class separation.

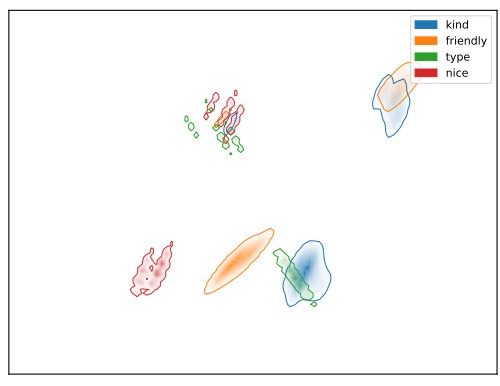

(c) Word with multiple meanings: `kind`.

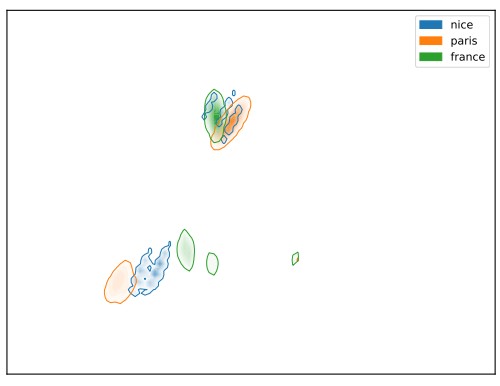

(d) Explaining a failed association: `nice`.

Figure 3: Directly visualizing high-dimensional word embeddings.

## 5 DISCUSSION AND CONCLUSION

Several characteristics determine the value and effectiveness of an embedding space for representation learning. The space must be large enough to embed a variety of metrics, while admitting a mathematical description compatible with learning algorithms; additional features, including direct interpretability, make it easier to understand, analyze, and potentially debug the output of a representation learning procedure. Based on their theoretical properties, Wasserstein spaces are strong candidates for representing complex semantic structures, when the capacity of Euclidean space does not suffice. Empirically, entropy-regularized Wasserstein distances are effective for embedding a wide variety of semantic structures, while enabling direct visualization of the embedding.

Our work suggests several directions for additional research. Beyond simple extensions like weighting points in the point cloud, one observation is that we can lift nearly *any* representation space $\mathcal{X}$ to distributions over that space $\mathcal{W}(\mathcal{X})$ represented as point clouds; in this paper we focused on the case $\mathcal{X} = \mathbb{R}^n$. Since $\mathcal{X}$ embeds within $\mathcal{W}(\mathcal{X})$ using $\delta$-functions, this might be viewed as a general "lifting" procedure increasing the capacity of a representation. We can also consider other tasks, such as co-embedding of different modalities into the same transport space. Additionally, our empirical results suggest that theoretical study of the embedding capacity of Sinkhorn divergences may be profitable. Finally, following recent work on computing geodesics in Wasserstein space (Seguy & Cuturi, 2015), it may be interesting to invert the learned mappings and use them for interpolation.

ACKNOWLEDGEMENTS

J. Solomon acknowledges the support of Army Research Office grant W911NF-12-R-0011 ("Smooth Modeling of Flows on Graphs"), of National Science Foundation grant IIS-1838071 ("BIGDATA:F: Statistical and Computational Optimal Transport for Geometric Data Analysis"), from an Amazon Research Award, from the MIT–IBM Watson AI Laboratory, and from the Skoltech–MIT Next Generation Program.

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
