# OpenReview forum: "Learning Embeddings into Entropic Wasserstein Spaces"
_ICLR.cc/2019/Conference_

### Official Review · AnonReviewer1 · 2018-10-29
**Very Limited Contribution**

**Rating:** 3
**Confidence:** 4

**Review:**

The paper proposes embedding the data into low-dimensional Wasserstein spaces. These spaces are larger and more flexible than Euclidean spaces and thus, can capture the underlying structure of the data more accurately. However, the paper simply uses the automatic differentiation to calculate the gradients. Thus, it offers almost no theoretical contribution (for instance, how to calculate these embeddings more efficiently (e.g. faster or more efficient calculation of the Sinkhorn divergence), how to motivate loss functions than can benefit the structure of the Wasserstein spaces, what the interpretation of phi(x) is for each problem, e. g. word embedding, etc.). Additionally, the experiments are unclear and need further improvement. For instance, which method is used to find the Euclidean embedding of the datasets? Have you tried any alternative loss functions for the discussed problems? Are these embeddings useful (classification accuracy, other distortion measure)? How does the 2-D visualizations compare to DR methods such as t-SNE? What is complexity of the method? How does the runtime compare to similar methods?

---

> ### Author Response · Authors · 2018-11-13
> **Thank you for your feedback!**
>
> Thank you for your comments! We are confident that the contributions of our work are both novel and interesting to the ICLR community.  We are eager to address your concerns, which can be addressed in a minor revision to our text or experiments.  Some responses are provided here, and we are happy to continue the conversation and/or provide additional information at your request.
>
> Most importantly, although you mention automatic differentiation as a negative, its use for approximating gradients of optimal transport-based divergences is non-obvious and was very recently a primary contribution of the AISTATS paper (Genevay 2018), which provides a way to differentiate transport for use in a neural network that is both stable and efficient. In a sense, our work is among the first to apply these new developments to a representation learning/embedding problem.
>
> More generally, efficient evaluation of optimal transport distances and their derivatives is a well-known and long-standing challenge in optimization. We leverage the current state of the art, in terms of efficiency, by using the Sinkhorn approximation to the Wasserstein distance (Section 2.2).
>
> While we appreciate the suggestion to show t-SNE plots and can add them to the paper if acceptance to ICLR is contingent on this change, it is worth noting that they will communicate different, more limited information in comparison to our visualizations. As is well-known in the data science community, it is easy to ascribe signal to noise when interpreting the locations of the t-SNE points, as they are not intended to capture locations or distances in the original embedding space. Of course, we are happy to generate the figures and see if they are useful, at your request.
>
> We respectfully highlight a few instances in which your questions are partially addressed in the existing text. We will gladly revise and/or augment the text for clarity, with guidance on the most effective ways to communicate the ideas below.
>
> 1. The interpretation of phi(x) for word embeddings is discussed in Section 4.2.1, where we demonstrate direct visualization of the embedding.
> 2. The Euclidean and hyperbolic embeddings are computed nearly identically to the Wasserstein embedding -- same loss function, same optimizer (Adam), different learning rate. This was mentioned in Section 4.1, fifth paragraph; we have also edited this to clarify that the optimizer is the same, and would be happy to add additional detail to this description.
> 3. The loss function for the first problem (complex networks) is dictated by the problem statement itself: We are establishing that one can learn Wasserstein embeddings that achieve low mean distortion, so our loss is the mean distortion. This is stated in Section 4.1, paragraph 2. Note that distortion is the standard criterion for metric embeddings.
> 4. The utility of the word embeddings for scoring semantic similarity of words is described in Section 4.2, paragraph 4, and in Table 2, where we evaluate the word embeddings on several benchmarks. Note also that we compare to five alternative methods.
>
> Thank you again for your feedback!  We appreciate your time and look forward to the possibility of sharing our work in ICLR soon.
>
> (Genevay 2018) Aude Genevay, Gabriel Peyré, Marco Cuturi. Learning Generative Models with Sinkhorn Divergences. AISTATS 2018.

---

> > ### Comment · AnonReviewer1 · 2018-11-21
> > **Thanks for providing further explanations**
> >
> > I am however still concerned with what your actual contribution is. As you mentioned in your feedback, you only apply the results of (Genevay 2018) and the Sinkhorn approximation on a number of problems. Your paper is more of a case study of Wasserstein embedding. Your work certainly has limited novelty and theoretical contribution.
> >
> > Thanks for clarifying some of the concerns in your feedback. Some of concerns are still not addressed:
> >
> > - Do you design any particular loss function that uses the Wasserstein structure? Seems like you are only considering the losses that have been used before for Euclidean embedding. In Wasserstein space, your instances are (non-negative) probability distributions (that sum up to one). How is your formulation (in e.g. Word2Cloud) any different than replacing the Euclidean distance with a Wasserstein distance in the loss function?
> >
> > - What the interpretation of phi(x) is for each problem, e. g. word embedding? You mention 4.2.1 in your feedback. But what does it mean to present a word as a distribution? Is it somehow a distribution over some latent topics, similar to LDA? Also, I feel like I am not totally understanding your point when you say you can directly visualize the Wasserstein embedding without using DR method. Can't you also directly visualize a 2-D embedding in a Euclidean space?
> >
> > " Whereas dimensionality reduction only approximately captures closeness of points in the embedding space, with Wasserstein embeddings we can see the exact embedding of each input, by visualizing the point cloud."
> > What does "exact" mean in this context?
> >
> > - Are these embeddings useful (classification accuracy, other distortion measure)? Do you have any experimental evidence for other measures?
> >
> > - What is complexity of the method? How does the runtime compare to Euclidean embedding? Can you also provide some runtime analysis?

---

> > > ### Author Response · Authors · 2018-11-27
> > > **Thank you for clarifying! (1/2)**
> > >
> > > Thank you for clarifying your remaining questions.  We are convinced of the novelty and interest of our contribution and are still hoping to demonstrate it to you.
> > >
> > > *** NOVELTY ***
> > >
> > > There seems to be some confusion regarding the particular contributions of our paper. We would characterize our main contributions as follows:
> > >
> > > 1. NEW REPRESENTATION:  We propose to learn embeddings of arbitrary inputs, such as images or words, as discrete probability distributions under an optimal transport metric. Notably, embedding into non-Euclidean spaces is currently an active area of research in the representation learning community (see for example (Nickel & Kiela 2017) and (Muzellec & Cuturi 2018)). To our knowledge, the idea of learning an embedding into a discrete transport metric has never been explored.
> > >
> > > 2.  EMBEDDING CAPACITY:  We show empirically that such learned embeddings can preserve a wide variety of metrics (Section 4.1), with lower distortion than baseline metrics. This is interesting both from the pragmatic perspective, that our learned embeddings are more flexible than the baselines, and also from the theoretical perspective, suggesting that the Sinkhorn divergence might inherit the favorable embedding properties of the Wasserstein distance identified in Section 2.3.
> > >
> > > 3.  WORD EMBEDDING:  We show empirically that useful Wasserstein embeddings can be learned for words (Section 4.2), achieving scores comparable to the state of the art on a set of semantic similarity benchmarks, which require predicting human similarity judgments.  This contribution extends a recent line of work on word embeddings that are more general than “a single point in a vector space.” See for example word2gauss (Vilnis & McCallum 2014) and Poincare embedding (Nickel & Kiela 2017), for other work in this direction.
> > >
> > > 4.  VISUALIZATION:  We demonstrate that Wasserstein embeddings over low-dimensional ground spaces can be directly visualized, unlike other high-dimensional embeddings (Section 4.2.1), and that, in the case of word embeddings, these visualizations reveal useful information for interpreting and debugging the embedding.
> > >
> > > Again, we are confident that these are novel and interesting ideas. Our ideas are well-tested empirically, are linked in the discussion to relevant existing theoretical results, and suggest new theoretical challenges that will inspire follow-on work. For recent papers with a similar empirical approach to proposing embeddings into alternative metric spaces, see for example (Nickel & Kiela 2017) and (Muzellec & Cuturi 2018).
> > >
> > > *** LOSS FUNCTION ***
> > >
> > > Regarding designing a loss function that uses the Wasserstein structure: It is not clear to us what this would entail. The message of our paper is that learning an embedding into an entropic Wasserstein space is both possible and useful, so our focus is on the impact of the Wasserstein metric and not the outer loss (which is problem-specific and agnostic to the particular divergence used, see Equation 7). This focus on the metric over the loss is very much in the vein of recent work, such as (Nickel & Kiela 2017) and (Muzellec & Cuturi 2018). Moreover, separating the target divergence from the loss function can be advantageous for learning embeddings, as it allows greater flexibility in designing the learning system than tailoring an ad hoc loss function for each task-representation pair. If you provide additional detail or an example as to what you mean by a loss function that “uses the Wasserstein structure,” we are happy to provide additional clarification.
> > >
> > > *** INTERPRETATION OF PHI(X) ***
> > >
> > > Regarding interpretation of phi(x) for word embeddings: Each word is embedded as a discrete distribution in an entropic Wasserstein space, so its interpretation requires thinking about the notion of nearness induced by the transport metric. Two distributions are near under a transport metric only if they are similar both in shape and location -- i.e. their supports are close and their densities are similar -- as differences in either can require transporting some of the probability mass over a large distance. This is what makes Wasserstein spaces flexible: We can encode semantics in both the shape and location of the distribution, providing an enormous embedding capacity even when the ground space (i.e. the support) is low-dimensional. Unlike a traditional mixture model (such as LDA), in which each location in the ground space (i.e. each component of the mixture) has a fixed semantic interpretation, in our case interpreting phi(x) requires looking at the entire embedded distribution, in the context of other embedded distributions. We illustrate how to do so in Section 4.2.1.

---

> > > > ### Author Response · Authors · 2018-11-27
> > > > **Thank you for clarifying! (2/2)**
> > > >
> > > > *** DIRECT VISUALIZATION ***
> > > >
> > > > Regarding the meaning of direct visualization: Keep in mind that an embedding into a discrete probability distribution is high-dimensional (i.e. it has many parameters, being the point locations), even when the ground space is low-dimensional. If a discrete distribution is supported at 8 points in a 2-D ground space, it has 8x2=16 parameters, so the most direct comparison would be to a 16-D Euclidean embedding, which cannot be visualized without dimensionality reduction. Yet in our case we do not need to apply dimensionality reduction, because the embedding is interpretable as a point cloud in the low-dimensional ground space. We can directly visualize this point cloud. This is the sense of the word “exact:” We need not apply any dimensionality reduction before visualizing the point cloud.
> > > >
> > > > *** UTILITY ***
> > > >
> > > > Regarding utility of the embedding: Please see Section 4.2, in which we show performance on a set of semantic similarity benchmarks, which require predicting human semantic similarity judgments. Note that in Section 4.1 we use the mean distortion as it is the standard measure for metric embeddings and directly characterizes the representational capacity of the embedding.
> > > >
> > > > *** RUNTIME ***
> > > >
> > > > Regarding runtime: Complexity of the Sinkhorn divergence computation is discussed in (Cuturi 2013) and (Soules 1991), with the latter showing linear convergence of the Sinkhorn iteration. Automatic differentiation of the Sinkhorn iteration has the same complexity. The runtime complexity of our proposed method is identical to that of (Genevay 2018).
> > > >
> > > >
> > > > (Nickel & Kiela 2017) Maximilian Nickel and Douwe Kiela. Poincare embeddings for learning hierarchical representations. In NIPS (2017).
> > > > (Muzellec & Cuturi 2018) Boris Muzellec and Marco Cuturi. Generalizing point embeddings using the Wasserstein space of elliptical distributions. In NIPS (2018).
> > > > (Vilnis & McCallum 2014), Luke Vilnis and Andrew McCallum. Word representations via Gaussian embedding. In ICLR (2015).
> > > > (Cuturi 2013) Marco Cuturi. Sinkhorn distances: lightspeed computation of optimal transport. In NIPS (2013).
> > > > (Soules 1991) George Soules. The rate of convergence of Sinkhorn balancing. Linear Algebra and its Applications 150 (1991).
> > > > (Genevay 2018) Aude Genevay, Gabriel Peyré, Marco Cuturi. Learning Generative Models with Sinkhorn Divergences. In AISTATS (2018).

---

> > > > > ### Comment · AnonReviewer1 · 2018-11-27
> > > > > **Thanks again for the clarifications**
> > > > >
> > > > > I carefully went through the paper and your answers. However, I am sorry to tell that I am still not convinced with the novelty of your work. Your paper should be presented as a case study of applying Wasserstein distances to the known problems and comparing to the Euclidean embeddings. However, the first 4 pages of your paper is simply some background review and your original work is presented afterwards on only two main problems. Your network embedding results on random networks are good. However, the improvement on real networks does not seem to be statistically significant. Additionally, I am still not convinced with your argument about 2-D visualizations. There is no novelty here in a sense that you can do the same visualization in a Euclidean space. Moreover, given that your axis are probability values, plotting the data in an orthogonal axis is probably not the best idea. The correct way of representing the data would be on a simplex.
> > > > >
> > > > > Overall, I would like to keep my initial score. However, I leave the final decision to the AC.

---

> > > > > > ### Author Response · Authors · 2018-11-28
> > > > > > **Some confusion**
> > > > > >
> > > > > > Thank you for responding. We are significantly confused by two of your statements:
> > > > > >
> > > > > > “... given that your axis are probability values, plotting the data in an orthogonal axis is probably not the best idea. The correct way of representing the data would be on a simplex.”
> > > > > >
> > > > > > This statement is false. As noted by Reviewer 2, our axes are **not** probability values. This is indeed the central concept of the paper. We are representing inputs as sums of Diracs and optimizing the locations of these Diracs (Section 3.2). As the Diracs are weighted uniformly, they can be visualized as point clouds in the ground metric space (which, in Section 4.2.1, is R^2). Again, this is fundamental to understanding what we did; it is the central concept of the paper.
> > > > > >
> > > > > > “... you can do the same visualization in a Euclidean space.”
> > > > > >
> > > > > > This statement is also false. A high-dimensional Euclidean embedding **cannot** be visualized without dimensionality reduction. A point cloud, meanwhile, can be directly visualized since it consists of (multiple) points in two or three dimensions.
> > > > > >
> > > > > > While we appreciate your willingness to respond, the issues above appear to indicate a fundamental misunderstanding of our submission.  While we still hope you might consider revising your score or confidence level, we also respect your suggestion to defer to the Area Chair.

---

### Official Review · AnonReviewer2 · 2018-10-30
**A very nice and original work.**

**Rating:** 7
**Confidence:** 4

**Review:**

The paper ‘Learning Discrete Wasserstein Embeddings' describes a new embedding method that,
contrary to usual embedding approaches, does not try to embed (complex, structured) data into an
Hilbertian space where Euclidean distance is used, but rather to the space of probability measures
endowed with the Wasserstein distance. As such, data are embed on an empirical
distribution supported by Diracs, which locations can be determined by a map that is learnt from data.
Interestingly, authors note a 'potential universality' for W_p(R^3) (from a result of Andoni et al., 2015),
suggesting that having Diracs in R^3 could embed potentially any kind of metric on symbolic data.

Experimental validations are presented on graph and word embedding, and a discussion on visualization of
the embedding is also proposed (since the Diracs are located in a low dimensional space).

All in all the paper is very clear and interesting. The idea of embedding in a Wasserstein space is
original (up to my knowledge) and well described. I definitely believe that this work should be presented
at ICLR. I have a couple of questions and remarks for the authors:
 - It is noted in section 3.2 that both Diracs location and associated weights could be optimized. Yet the authors
   chose to only optimize locations. Why not only optimizing the weights (as in an Eulerian view of probability
   distributions) ? The sentence involving works of Brancolini and Claici 2018 is not clear to me. Why weighting
   does not improve asymptotically the approximation quality ?
 - Introducing the entropic regularization is mainly done for being able to differentiate the Wasserstein
   distance. However, few is said on the embeddability of metrics in W^\lambda_p(R). Is using an entropic
   version of W moderating the capacity of embedding ? At least experimentally, a discussion could be made
   on the choice of the regularization parameter, at least in section 4.1. In eq. (9), it seems that it is not
   the regularized version of W. ?
 - I assume that the mapping is hard to invert, but did the authors tried to experiment reconstructing an object
   of interest by following a geodesic in the Wasserstein space ?
 - It seems to me that authors never give generalization results. What is the performance of the metric approximation
   when tested on unseen graphs or words ? This point should be clarified in the experiment.

---

> ### Author Response · Authors · 2018-11-13
> **Thank you for your comments!**
>
> Thank you for your thoughtful comments! We have made several small edits, according to your suggestions.
>
> (Brancolini 2009) and (Kloeckner 2012) show that, when using a weighted point cloud to approximate an absolutely continuous, compactly-supported measure, the order of convergence in p-Wasserstein distance when allowing non-uniform weights is O(n^(-1/d)), which is the same rate as when restricted to uniform weights (Dudley 1969). Non-uniform weights might buy you an improved constant term, but the rate is the same. We have expanded the description of this fact in Section 3.2, for clarity.
>
> The embedding capacity of W^\lambda_p(R^d) is unknown, so far as we are aware, except in the weak sense that the approximation error with respect to the p-Wasserstein distance vanishes as the regularizer is taken zero (Carlier 2017; Genevay 2018). We have added a discussion of this distinction between W_p and W^\lambda_p to Section 2.3.
>
> Inverting the mapping and following geodesics in Wasserstein space would definitely be interesting. We have added this to the suggested future work in the conclusion. An approach such as (Seguy 2015) might be useful here.
>
> We have updated the notation in eq. (9) to highlight the fact that entropic regularization is used for learning word embeddings.
>
> We have added a comment on generalization performance to Section 4.1, paragraph 3.
>
> Thank you again for your feedback!
>
> (Brancolini 2009) Alessio Brancolini, Giuseppe Buttazzo, Filippo Santambrogio, Eugene Stepanov. Long-term planning versus short-term planning in the asymptotical location problem. ESAIM: Control, Optimisation, and Calculus of Variations 15, no. 3 (2009).
> (Kloeckner 2012) Benoit Kloeckner. Approximation by Finitely Supported Measures. ESAIM: Control, Optimisation, and Calculus of Variations 18, no. 2 (2012).
> (Dudley 1969) Richard Dudley. The Speed of Mean Glivenko-Cantelli Convergence. Annals of Mathematical Statistics 40, no. 1 (1969).
> (Carlier 2017) Guillaume Carlier, Vincent Duval, Gabriel Peyré, Bernhard Schmitzer. Convergence of Entropic Schemes for Optimal Transport and Gradient Flows. SIAM Journal on Mathematical Analysis 49, no. 2 (2017).
> (Genevay 2018) Aude Genevay, Lénaic Chizat, Francis Bach, Marco Cuturi, Gabriel Peyré. Sample Complexity of Sinkhorn Divergences. arXiv:1810.02733 (2018).
> (Seguy 2015) Vivien Seguy and Marco Cuturi. Principal Geodesic Analysis for Probability Measures under the Optimal Transport Metric. NIPS 2015.

---

> > ### Comment · AnonReviewer2 · 2018-11-21
> > **About generalization performance**
> >
> > Thanks for the detailed explanation. I am somehow disappointed by the note you added about generalization. While I understand that your main concern is the representational capacity of the model, does it mean that the learnt embedding can not be applied to unseen data ? It might potentially lower the possibilities of doing geodesic analysis in the embedding space, as discussed in the perspectives, and raise also concerns about overfitting to the pairs of examples seen in the learning set.

---

> > > ### Author Response · Authors · 2018-11-27
> > > **Regarding generalization**
> > >
> > > Thanks for getting back to us!  We certainly do not want to disappoint and are enthusiastic to change your mind before the discussion period ends.
> > >
> > > The note about generalization in Section 4.1 is specific to the experiments in that section; other parts of the paper address generalization directly. Generalization of the learned models to unseen input pairs and new tasks is certainly possible, and this is in fact one purpose of the discussion in Section 4.2. In particular, we demonstrate generalization of the learned word embeddings on an independent prediction task, in which we successfully predict human semantic similarity judgments. The positive results in Table 2 indicate that such generalization is indeed possible.
> > >
> > > In general, the question of generalization of learned embeddings is an interesting one and we expect this to be an important area for application and further development of the proposed method.  If there are experiments you wish for us to include in our camera-ready draft to demonstrate generalization in our existing framework, we would be more than happy to include them.
> > >
> > > We appreciate your feedback.  Please let us know if you have further concerns!

---

### Official Review · AnonReviewer3 · 2018-11-02
**A simple and interesting idea on how to map data in a discrete (Wasserstein) space**

**Rating:** 7
**Confidence:** 3

**Review:**

This paper learns embeddings in a discrete space of probability distributions, endowed with a (regularized) Wasserstein distance.

pros:

- interesting idea, nice results, mostly readable presentation.
- the paper is mostly experimental but the message delivers clearly the paper’s objective
- the direct visualisation is interesting
- the paper suggests interesting problems related to the technique

cons:

- to be fair with the technique, the title should mention the fact that the paper minimises a regularised version of Wasserstein distances (Wasserstein -> Sinkhorn ? put “regularised" ?)
- and to be fair, the paper should put some warnings related to regularisation -- this is not a distance anymore, sparsity is affected by regularisation (which may affect visualisation). Put some reminders in the conclusion, reword at least the third paragraph in the introduction.
- the paper could have been a little bit more detailed on Section 2.3, in particular for its third paragraph. Even when it is an experimental paper.
- the direct visualisation is interesting in the general case but has in fact a problem when distributions are highly multimodal, which can be the case in NLP. This blurs the interpretation.
- the paper delivers a superficial message on the representation: I do not consider that nice having modes near physical locations (Paris, France) is wrong. It is also a city. However, it would have been interesting to see the modes of “city” (or similar) to check whether the system indeed did something semantically wrong.

Questions:

- beyond that last remark comes the problem as to whether one can ensure that semantic hierarchies appear in the plot: for example if Nice was only a city, would we observe a minimal intersection with the support of word “city” ? (intersection to be understood at minimal level set, not necessarily 0).

---

> ### Author Response · Authors · 2018-11-13
> **Thank you for your feedback!**
>
> Thank you for your helpful comments! We made several small revisions, according to your suggestions.
>
> To make it more clear that we are using regularized transport, we have made four changes:
> 1.  The title is now “Learning Entropic Wasserstein Embeddings.”
> 2.  In the introduction, we expanded the existing discussion of the Sinkhorn divergence and moved it to a separate paragraph (now the fifth paragraph).
> 3.  In Section 2.3, we note that we are using the Sinkhorn divergence, and briefly discuss its theoretical properties.
> 4.  In the conclusion, we state that our empirical results are for the Sinkhorn divergence.
>
> We have expanded the third paragraph of Section 2.3, to provide more detail on embedding capacity results for Wasserstein spaces.
>
> Our intent with including “nice” in the visualization section was to show first an error (Figure 3c), which is that “nice” is visibly distant from “kind” (a synonym), then to show the explanation for the error (Figure 3d), which is that the network learned that “nice” is a city in France, while ignoring its second meaning. To make this clearer, we have changed the caption for Figure 3d to “Explaining a failed association.” Please let us know if this addresses your comment.
>
> The idea to look at semantic hierarchies is very interesting, and we did investigate this briefly. We observed that the minimal level sets of the parent and child in the hierarchy were often partially overlapping -- for instance the embedding of “city” has two major modes, one of which overlaps with a number of cities, including “nice.” As you point out, however, it is not immediately clear how hierarchies should manifest in this type of embedding, particularly as the parent can have semantics not shared by the children (e.g. “state,” which has multiple non-geographic meanings). Perhaps the parent (or one of its modes) should be near to the barycenter of the children? This would be interesting to investigate further.
>
> Thank you again for your feedback!

---

### Public Comment · (anonymous) · 2018-11-19
**Questions regarding word2cloud**

1. In eq. 9, should (1-r_{x_i,x_j}) and r_{x_i,x_j} be interchanged? When r_{x_i,x_j}=1, we should want to minimize the Wasserstein distance between the related embeddings. And when r_{x_i,x_j}=0, we would want to have the unrelated embeddings at a margin of at least m. For context, in the paper by Hadsell et al which is cited as the source of the contrastive loss function, the binary relationship variable is defined in the opposite manner: "Y = 0 if X1 and X2 are deemed similar, and Y = 1 if they are deemed dissimilar."

2. What does the point cloud correspond to? I couldn't find a clear statement of it. My best guess was that it is the set of all point embeddings associated with words in a sentence. This discrete point cloud is then made continuous by KDE. Is my understanding correct?

Thanks for your time.

---

> ### Author Response · Authors · 2018-11-20
> **Thanks!**
>
> Thanks for commenting!
>
> 1.  Thanks for noticing the typo in Eq. (9). It’s been corrected.
> 2.  Each word is embedded as a point cloud (i.e. a uniformly weighted discrete distribution). In Section 4.2.1, we apply KDE to each point cloud (i.e. word), separate from the others.
>
> Let us know if you have any other questions.

---

### Public Comment · (anonymous) · 2018-12-28
**Details of the Siamese architecture for word experiments**

Interesting work.

- Could you please give more specific details about the Siamese architecture? In particular, how do you represent the word x_i at the input?  Also, what is the point-cloud embedding layer precisely?

- I guess the first term of eq 9 (contrastive loss), should be squared.

Thank you!

---

> ### Author Response · Authors · 2019-01-08
> **Siamese architecture**
>
> Thanks for the comment.
>
> - The input to each branch of our Siamese architecture is a one-hot encoding of a word from the vocabulary. Next, each of the branches goes through a fully-connected linear layer, followed by the point cloud layer, which is a fully-connected linear layer whose output is normalized to lie within the unit ball. The Sinkhorn distance is then applied to the outputs of the two branches to compute the distances used within the contrastive loss.
>
> - You’re right, thanks!

---

### Public Comment · ~Sidak_Pal_Singh1 · 2019-01-15
**Regarding work on embedding with discrete distributions in Wasserstein space**

Hi,

This is indeed a well written paper and gives an interesting perspective on low distortion embeddings into Wasserstein space.

I wanted to point out that our work https://arxiv.org/abs/1808.09663 (and the openreview version from june last year https://openreview.net/pdf?id=Bkx2jd4Nx7), has considered representing entities (such as words, sentences, etc) as (discrete) distribution over their contexts, where the contexts are essentially point embeddings in a low-dimensional space. We use optimal transport/Wasserstein distance to compare & compose entities and this is approximated via the entropic regularization/Sinkhorn divergence (Cuturi, 2013).

It would be great if you could maybe update your paper to reflect this and include a reference to our work.

Thanks for your time.

Best,
Sidak

---

### Meta-Review · Area_Chair1 · 2018-12-13
**An interesting word embedding method**

**Confidence:** 4
**Recommendation:** Accept (Poster)

**Metareview:**


+ An interesting and original idea of embedding words into the (very low dimensional) Wasserstein space, i.e. clouds of points in a low-dimensional space
+ As the space is low-dimensional (2D), it can be directly visualized.
+ I could imagine the technique to be useful in social / human science for data visualization, the visualization is more faithful to what the model is doing than t-SNE plots of high-dimensional embeddings
+ Though not the first method to embed words as densities but seemingly the first one which shows that multi-modality  / multiple senses are captured (except for models which capture discrete senses)
+ The paper is very well written

-  The results are not very convincing but show that embeddings do capture word similarity (even when training the model on a small dataset)
-  The approach is not very scalable (hence evaluation on 17M corpus)
-  The method cannot be used to deal with data sparsity, though (very) interesting for visualization
-  This is mostly an empirical paper (i.e. an interesting application of an existing method)

The reviewers are split. One reviewer is negative as they are unclear what the technical contribution is (but seems a bit biased against empirical papers). Another two find the paper very interesting.